# A Direct Comparison of the Relationship of Epigenetic Aging and Epigenetic Substance Consumption Markers to Mortality in the Framingham Heart Study

**DOI:** 10.3390/genes10010051

**Published:** 2019-01-15

**Authors:** James A. Mills, Steven R.H. Beach, Meeshanthini Dogan, Ron L. Simons, Frederick X. Gibbons, Jeffrey D. Long, Robert Philibert

**Affiliations:** 1Department of Psychiatry, University of Iowa, Iowa City, IA 52242, USA; Jim-mills@uiowa.edu (J.A.M.); Jeffrey-long@uiowa.edu (J.D.L.); 2Center for Family Research, University of Georgia, Athens, GA 30602, USA; srhbeach@uga.edu (S.R.H.B.); ron.lee.simons@gmail.com (R.L.S.); 3Cardio Diagnostics LLC, Coralville, IA 52241, USA; mdogan@bdmethylation.com; 4Department of Psychology, University of Connecticut, Stoors, CT 06268, USA; rick.gibbons@uconn.edu; 5Department of Biostatistics, University of Iowa, Iowa City, IA 52242, USA; 6Behavioral Diagnostics LLC, Coralville, IA 52241, USA

**Keywords:** epigenetic aging, alcohol, smoking, DNA methylation, aryl hydrocarbon receptor repressor, survival

## Abstract

A number of studies have examined the relationship of indices of epigenetic aging (EA) to key health outcomes. Unfortunately, our understanding of the relationship of EA to mortality and substance use-related health variables is unclear. In order to clarify these interpretations, we analyzed the relationship of the Levine EA index (LEA), as well as established epigenetic indices of cigarette (cg05575921) and alcohol consumption (cg04987734), to all-cause mortality in the Framingham Heart Study Offspring Cohort (*n* = 2256) Cox proportional hazards regression. We found that cg05575921 and cg04987734 had an independent effect relative to LEA and vice versa, with the model including all the predictors having better performance than models with either LEA or cg05575921 and cg04987734 alone. After correction for multiple comparisons, 195 and 327, respectively, of the 513 markers in the LEA index, as well as the overall index itself, were significantly associated with cg05575921 and cg04987734 methylation status. We conclude that the epigenetic indices of substance use have an independent effect over and above LEA, and are slightly stronger predictors of mortality in head-to-head comparisons. We also conclude that the majority of the strength of association conveyed by the LEA is secondary to smoking and drinking behaviors, and that efforts to promote healthy aging should continue to focus on addressing substance use.

## 1. Introduction

Understanding the mechanisms through which broadly defined lifestyle factors result in premature morbidity and mortality is critical to the formulation of cost-effective strategies to promote healthy aging. In efforts to pinpoint these mechanisms, epidemiologists have examined large longitudinally characterized cohorts in order to identify key environmental risks or behaviors that are associated with disease. Although many of these study cohorts have made substantial contributions, one particularly influential population has been the Framingham Heart Study (FHS). Founded in 1948, the FHS now includes data from over 20,000 individuals and has been used to generate over 3600 manuscripts detailing the relationship of key risk variables to not only heart disease, but also cancer, depression and normative aging [1].

One sub-group of the FHS that has been extensively studied with respect to mortality is the Offspring Cohort [2]. The Offspring Cohort was initiated in 1971, with the 5124 adult children (and spouses) of the original subjects each having participated in at least one of the 9 waves of characterizations. Using DNA collected during Wave 8 (~2008), genome wide assessments of methylation for a large number of the Offspring Cohort were performed. Not surprisingly, these valuable epigenetic data have been used extensively by a number of investigators including those seeking to understand the relationship of changes in DNA methylation to key health outcomes.

In particular, the genome wide epigenetic data from this FHS cohort have been used to test or construct various indices of epigenetic aging [3,4,5,6,7]. These indices, each of which use methylation data from a large number of loci, can be used to accurately infer chronological age. The difference between chronological age and inferred age, termed Accelerated Aging (AA), has been associated with a wide variety of health outcomes. Unfortunately, interpretation of the changes in these indices to health outcomes is often difficult because of the multitude of markers used in age imputation. Still, by clarifying the mechanisms through which methylation at these loci are altered, we could gain new insight for promoting longevity.

Two indices for calculating AA, one from Horvath and associates, and the other from Brenner and Colleagues, have garnered recent attention. The first, referred to as the Levine Aging Index (LEA), is the 513 marker panel recently described by Horvath and colleagues (*2018*), *that “strongly outperforms previous measures in regards to predictions for a variety of aging outcomes, including all-cause mortality.” [8]. Interestingly, although the Horvath group also shows strong relationships of the LEA to a variety of important outcomes associated with smoking including lung cancer, they* state that they do not find a robust connection between their index and their chosen index of smoking, self-reported pack year consumption nor smoking in general [8]. This is noteworthy because in 2016, Brenner and colleagues used data from the ESTHER (Epidemiologische Studie zu Chancen der Verhütung, Früherkennung und optimierten THerapie chronischer ERkrankungen) and KORA (Kooperative Gesundheitsforschung in der Region Augsburg) cohorts to construct an all-cause mortality risk (Brenner mortality risk; BMR) clock with 48 of the top 58 CpGs, most prominently cg05575921, in the BMR index being associated with smoking [9]. What is more, Brenner and colleagues found little additional contributions of the Hannum aging index or a prior version of the Horvath clock on mortality in their cohorts to the ability of their smoking driven clock to predict all-cause mortality [9]. Most recently, Brenner and colleagues extended these prior findings and showed the usefulness of the BMR clock, and a novel Frailty Index (FI) [10], which is comprised of 34 measures of deficits (symptoms, signs, diseases or limitations in activities of daily living), to predict survival in the ESTHER cohort [11]. Crucially, they find no additional predictive value from the earlier Horvath clock after controlling for the effects of BMR and FI. Even more surprisingly, they find that three CpG residues in their BMR clock, most notably cg04987734, are also independently associated with the FI index.

The discrepancies in the findings from two highly regarded groups are puzzling for two reasons. First, there is a marked dissonance in the perceived environmental drivers in the two clocks. Brenner and associates make it clear that smoking is the major factor behind the epigenetic changes in EA whereas Horvath and colleagues do not [8,9]. Second, Brenner and colleagues note the independent contribution of FI, which is in part driven by changes in cg04987734 methylation, to mortality [11].

Understanding the reasons for these differences regarding the effect of smoking on mortality and the biology of this new FI index could be of critical importance to the construction of effective public health measures for encouraging healthy aging. Indeed, if the new EA clocks can identify new potentially preventable causes of illness, these indices can make a monumental improvement to quality of life. Conversely, if they are failing to identify the main drivers of preventable causes of mortality and are detracting from current public health efforts, the diversion of investigative efforts and resources could have serious adverse consequences on the welfare of a vulnerable population. According to the Centers for Disease Control, smoking and excessive alcohol consumption are the first and third most common causes of preventable morbidity and mortality and national policies for prevention reflect those conclusions [12,13]. Do the recent findings suggest the need for change? Unfortunately, the FHS, which constituted over 1/3 in the test population used by Levine and colleagues, does not have cotinine levels, the most commonly used serological marker of smoking [14], at the Wave 8 time point to aid with an objective analysis of the relationship of smoking to the LEA.

Nevertheless, the second finding by Brenner mentioned above may be a key to resolving this issue and adding interpretability to these indices. Whereas it is generally appreciated that cg05575921 is a sensitive and specific indicator of smoking [15,16,17], it is generally not appreciated that cg04987734 is a sensitive and specific indicator of alcohol consumption. Since its original identification in 2014 [18], two studies showed that cg04987734 methylation is not influenced by smoking. The first study was by Liu and associates who used regression of self-reported smoking status to control for the effects of smoking at this locus [19]. The second was by our group in which we used only serum confirmed non-smoking alcohol cases in a subset of the analyses [10]. Remarkably, these two markers are quite capable of measuring smoking or drinking behaviors by themselves. In two separate studies, we have shown that cg05575921 methylation, as measured by the Illumina array or droplet digital polymerase chain reaction (PCR) has a Receiver Operating Characteristic (ROC) area under the curve (AUC) of 0.99 for classifying current smokers from lifetime non-smokers [17,20]. Similarly, in a single study we have shown that the AUC of the Illumina array measurement of cg04987734 methylation in classifying individuals with respect to heavy alcohol consumption from non-drinkers is 0.88 [10]. By using these epigenetic markers of smoking and drinking in tandem with indices of EA, we still may more exactly reveal the extent to which EA predicts mortality independently of smoking and/or drinking, thereby creating new insights into potentially preventable disease.

In this communication, we analyze the relationship of the Levin EA clock as well-established epigenetic markers of cigarette consumption (cg05575921) [15,16] and a newer epigenetic marker of alcohol consumption (cg04987734) [10,19,21] to survival (or all-cause survival) in the FHS Offspring Cohort.

## 2. Materials and Methods 

The data used in this study are from the Framingham Heart Study (FHS) and were previously prepared for epigenetic studies of smoking and heart disease [22,23]. All of the procedures and protocols used in this study were approved by the University of Iowa Institutional Review Board (IRB 201503802).

### 2.1. DNA Methylation Data

The clinical and epigenetic data used in this study were extracted from a larger dataset of 2295 individuals from the Framingham Heart Study (FHS). A complete description of the procedures used to prepare these data has been previously reported [22,23].

### 2.2. Clinical Assessments of the FHS Subjects

The clinical information used in this study was extracted from the Wave 8 examination report with additional survival data based on the Wave 9 survey. These clinical and demographic assessments include age, sex, self-reported current and past smoking status, coronary heart disease (CHD) and stroke status as determined by the FHS review panel, self-reported chronic obstructive pulmonary disease status (COPD), self-reported diabetes status, investigator clinical impression assessment of dementia status, systolic and diastolic blood pressure measurements, date of death (for 288 participants with death certificates) and dates for Wave 9 assessments, for those who participated. The data from those 17 individuals for whom death status was unknown and did not participate in Wave 9 analyses were excluded from the analyses.

### 2.3. Data Analysis

The LEA was calculated for each participant using a linear combination of 513 DNA methylation values as described by Levine et al [8]. The values for cg05575921 and cg04987734 were used directly without further transformation.

Initial group comparisons were performed at Wave 8, with comparisons of continuous variables evaluated with *t*-tests, and comparisons of categorical variables conducted using Fisher’s Exact Test [24]. 

The primary analyses were conducted using Cox proportional hazards regression [25]. The time metric was days from the Wave 8 visit (time 0) to death (all causes) or censoring (87% of the sample was censored). All the predictors in the proportional hazards models were measured at the time 0 baseline of the Wave 8 visit. Continuous variables were expressed in standard deviation (SD) units after scaling by using the overall mean and SD for each measure. Predictors were first considered individually in univariate models and then together in a series of multivariate models with different numbers of predictors, as described below. Harrell’s *C* was used to index the overall prediction accuracy of each model and pseudo R^2^ was used to quantify the strength of association of the model predictors with time to death [26,27]. Two other measures assessing model performance were also calculated, the integrated discrimination improvement (IDI) and the net reclassification improvement (NRI) [28]. In the base model (Model 1), only age and gender were used to predict all-cause mortality. In Models 2–9, we added known predictors of mortality (CHD, COPD, diabetes, stroke, and dementia), LEA, cg05575921, and cg04987734 one by one to the base model, which allowed us to examine changes in prediction accuracy and strength of association with time to death. We considered combinations of the methylation-based measures in Models 10 and 11. In Model 12 we included all of the commonly used predictor variables (CHD, COPD, diabetes, stroke, and dementia). In Model 13 we included LEA, in Model 14 cg05575921, and in Model 15 cg05575921 and cg04987734 in order to observe changes in concordance and model fit corresponding with the addition of the methylation-based measures. Finally, all predictors were included in Model 16. Hazard ratios with 95% confidence intervals are presented for all univariate models and the full multivariate model (Model 16).

In a final set of analyses, we correlated the relationship of smoking and drinking intensity as indicated by cg05575921 and cg04987734 methylation, respectively, with LEA and each of the methylation markers therein. Because of the large number of correlations, the *p*-values were adjusted for multiple comparisons using the False Discovery Rate (FDR) approach, which yielded *q*-values with significance defined as *q* ≤ 0.05 [29]. All analyses were conducted using R Version 3.5.1.

## 3. Results

The clinical and demographic characteristics of the 2256 individuals who contributed data to this study are detailed in Table 1. Each of the participants was White with an average age of 66 ± 9 years of age. A little more than half of the sample was female (55%), and 8% and 9%, respectively, of all participants reporting that they were currently smoking or had previously smoked. Coronary heart disease was present for 14% of the participants. A further 2% and 12%, respectively, reported that they had been diagnosed with COPD or diabetes, while nearly 5% reported a stroke prior to the Wave 8 interview. Notably, whereas the majority of objective measures were fairly equal between the genders, males were more likely to have CHD and diabetes (χ^2^ = 44.4, *p* < 0.0001 and χ^2^ = 7.7, *p* = 0.0055, respectively).

A total of 288 participants died (13%) between the Wave 8 assessment and attempted follow-up at Wave 9. During this period, the first individual died only 17 days after Wave 8 assessment and the last of the 288 participants died 3503 days after Wave 8 assessment.

Although cotinine testing was not conducted during Wave 8, the determination of DNA methylation status at cg05575921 and cg04987734 permits some objective insights into the smoking and drinking habits of the sample. Overall, cg05575921 methylation was significantly lower (*t* = −3.7, *p* = 0.0003) and cg04987734 methylation was significantly higher (*t* = 9.0, *p* < 0.0001) in males as compared to the females. Since demethylation of cg05575921 and hypermethylation of cg04987734 are indicative of higher levels of consumption of cigarettes and alcohol, respectively, and there is no difference between the genders in the methylation set points at these loci [10,20], this indicates that on average, the males of this study smoke and drank more heavily than their female counterparts.

Figure 1 and Figure 2 illustrate the distribution of cg05575921 and cg04987734 as determined by the Illumina 450 K array in this sample. Increasing cigarette consumption is associated with decreased DNA methylation at cg05575921 [15,16]. Inferring from previously published work using the 450 K array on participants who were biochemically verified lifetime non-smokers [17], the vast majority of those individuals with DNA methylation values above 0.8 are lifetime non-smokers. Though the exact cut point for cg05575921 status will vary from methylation array to methylation array, lower values at cg05575921 are associated with increased cigarette consumption with those individuals with methylation levels below 0.6 generally being heavy smokers (i.e., >20 cigarettes per day). Conversely, cg04987734 methylation is lower among abstinent individuals with increasing consumption of alcohol being steadily associated with increased methylation at this locus.

Table 2 presents hazard ratios with 95% confidence intervals from univariate Cox proportional hazards models for each of the clinical and epigenetic measures. Increased age and presence of CHD, COPD, diabetes, stroke, and dementia were all strongly associated with a higher risk of mortality. Increased epigenetic age was also found to be associated with an increase in the risk of death. DNA methylation values at the two loci for cigarette and alcohol consumption were both significantly associated with mortality. Decreased methylation at cg05575921 and increased methylation at cg04987734 were associated with a decrease in survival time. 

In order to examine the relationship between each of the putative risk variables and survival, we conducted proportional hazards regression analyses using sixteen separate models, calculating Harrell’s C, pseudo R^2^, IDI, and NRI for each (Table 3). In Model 1, only age and gender were used to predict the timing of death. In Models 2–6, we added single variables known to be highly predictive of mortality (CHD, COPD, diabetes, stroke, and dementia) to the base model and saw improvements in accuracy and model fit. For Model 7 we included LEA and found that it also improves accuracy and fit over the base model (C = 0.760; pseudo R^2^ = 0.119; IDI = 0.0149; NRI= 0.272). We added the alcohol and smoking methylation markers in Models 8–10 and observed that Model 9, with cg05575921, and Model 10, with cg05575921 and cg04987734, both generally had higher values for the model fit indices (C = 0.774 and 0.779; pseudo R^2^ = 0.130 and 0.135; IDI = 0.0290 and 0.0391; NRI = 0.237 and 0.264, respectively) than Models 2–7. Thus, Models 9 and 10 have stronger predictive accuracy than Models 2–7. In Model 11, LEA was included along with cg05575921 and cg04987734, resulting in an improvement over Model 10 (C = 0.787; pseudo R^2^ = 0.142; IDI = 0.0473; NRI = 0.313). Finally, we included all of the commonly used predictor variables, first without any of the methylation measures (Model 12), then with LEA (Model 13), cg05575921 (Model 14), and cg05575921 and cg04987734 (Model 15). The addition of the smoking and alcohol methylation markers in Model 15 resulted in improved prediction accuracy and stronger association with mortality (C = 0.806; pseudo R^2^ = 0.167; IDI = 0.0836; NRI = 0.329) in comparison to Model 12 (C = 0.779; pseudo R^2^ = 0.139; IDI = 0.0352; NRI = 0.277) and Model 13 (C = 0.788; pseudo R^2^ = 0.151; IDI = 0.0578; NRI = 0.286). The full model, Model 16, had the highest accuracy and fit statistics for all models considered (C = 0.810; pseudo R^2^ = 0.173; IDI = 0.0946; NRI = 0.358). Re-running Models 12 through 16 while excluding the data with respect to dementia status had a negligible effect on outcomes.

Table 4 lists the hazard ratio estimates for all predictors in Model 16. In this full model, the presence of CHD, COPD, and stroke remain highly predictive of time to death after adjustment for the other predictors (CHD Hazard Ratio (HR) 1.85, 95% CI = 1.412.42; COPD HR = 3.25, 95% CI = 2.095.07; Stroke HR = 3.10, 95% CI = 2.19–4.39). LEA was also predictive of mortality, as increased values were associated with increased risk of death (HR = 1.44, 95% CI = 1.20–1.72). Decreased methylation at cg05575921 and increased methylation at cg04987734 both remain highly predictive as well. Lower levels in baseline fractional methylation at cg05575921 and higher levels in baseline fractional methylation at cg04987734 result in an increased risk of mortality (HR = 0.71, 95% CI = 0.64–0.79 and HR = 1.21, and 95% CI = 1.10–1.34, respectively).

In Figure 3, we present the correlations of each clinical variable used in the models, as well as categorical variables of interest such as self-reported smoking. Age is significantly correlated with every variable except for gender. In the analyses, not surprisingly, LEA is high correlated with age (*r* = 0.77) with several other of the significant (*p* < 0.0001) correlations being noted with CHD (*r* = 0.20), diabetes (*r* = 0.14) and stroke (*r* = 0.13) being similar in magnitude to that of age alone (correlations of 0.18, 0.11 and 0.15, respectively). In contrast, age is only modestly correlated with cg05575921 methylation (*r* = 0.05, *p* < 0.007) but is more strongly correlated with cg04987734 methylation status (*r* = 0.20, *p* < 0.0001). As one would expect for a biomarker of smoking, cg05575921 is highly, but imperfectly, correlated with both former and current self-reported smoking. Finally, LEA is strongly associated with both cg05575921 and cg04987734 methylation (*p* < 0.0001).

In order to more exactly understand the significant relationship of the LEA to the smoking and drinking measures, we examined Pearson correlations between cg05575921 and cg04987734 methylation with each of the 513 methylation markers in LEA. At the nominal level, LEA and 235 of the 513 markers in the study were associated with cg05575921 methylation status while 344 of markers in the LEA index were significantly associated with cg04987734 status. After FDR adjustment, 195 and 327 of these markers, respectively, in the LEA index remain significantly associated with cg05575921 and cg04987734 methylation with the largest *q*-values exceeding 1 × 10^−26^ and 1 × 10^−160^, respectively (see Appendix A).

## 4. Discussion

In summary, we found that (1)cg05575921 and cg04987734 have predictive effects over and above LEA, but the converse is also true;(2)in the full model with all predictors, cg05575921 has the strongest standardized effect followed by LEA and cg04987734;(3)after adjustment for multiple comparisons, 38% and 64% of the markers in the LEA index are significantly associated with the objective epigenetic biomarkers of smoking and drinking (cg05575921 and cg04987734), respectively.


Limitations of the study include the fact that the FHS is a White, late middle-aged population and we did not include other important causes of death, such as cancer, in the final model.

The current data confirm and extend the prior findings by Brenner and colleagues [9,11] and show the value of using epigenetic biomarkers of both smoking and drinking to understand the effects of substance use on survival in the elderly. The evidence that these single markers accurately denote substance use is substantial. Dozens of genome wide studies have demonstrated the relationship of sensitive and specific relationship of demethylation at cg05575921 to smoking [15,16]. The data to support the use of cg04987734 to indicate alcohol consumption are not as extensive but three consecutive genome wide analyses, including one that used data from the FHS, have shown its relationship to drinking [10,19,21]. It is important to note that methylation changes at cg05575921 and cg04987734 also revert as functions of abstinence from smoking [30,31] and drinking [18], respectively, mirroring the improvements in life expectancy seen in those achieving abstinence. This reversibility of substance use-induced methylation changes is a potential boon to health care researchers and underscore the reasons why pack year metrics used in prior analyses may be poor choices for understanding the relationship of epigenetic indices to smoking. If a patient quits smoking or quits drinking excessively, the methylation signatures at these loci revert. Hence, even if the recall of a subject is accurate, the effects of the distant past may not be relevant. What is more, if the relationship of these changes in methylation in these two can be linked to changes in quality of life or survival, it is possible that investigators seeking to understand the cost benefit relationships of substance use interventions may be able to use changes of these in methylation as proxies of benefit or reduction of harm.

In this communication, we present four different metrics, Harrell’s C, pseudo R^2^, IDI and NRI, through which to analyze the effect of cg05575921, cg04987734, and LEA on survival prediction. To be clear, each of those metrics have strengths and weaknesses that are well noted in the literature [28,32,33,34]. Still, in each of those analyses, although the amount of additional information was rather small, LEA did seem to predict a small proportion of survival independent of the effects of cg05575921 and cg04987734. However, that small amount should not be construed as the limit of what can be estimated. Indeed, even our own studies of epigenetic biomarkers for cardiovascular disease indicate that epigenetic information independent of smoking and drinking can be harvested to derive sensitive and specific predictors of disease [23]. But using genome wide approaches may not be the best or only way to measure these predictors. Because of limitations of genome wide hybridization array in assessing methylation [35], once an environmental signal is detected, alternative mechanisms of assessing DNA methylation, such as methylation sensitive digital PCR, may be more effective mechanisms in capturing the epigenetic signal associated with an environmental factor [10,20]. This is particularly true for cg04987734, where the variance captured by digital PCR measurement of that locus is >30% greater than that of the array based signal [10].

A major advantage of the cg05575921 and cg04987734 methylation signals used in this study is interpretability. Dozens of case and control studies have shown that demethylation of the *aryl hydrocarbon receptor repressor* (AHRR) locus is a specific indicator of smoking and that it is not affected by alcohol consumption [15,16]. Conversely, although the data for cg04987734 is less extensive, to date, altered methylation at this locus appears specific to alcohol consumption and is not affected by smoking [10,19,21]. Still, despite the strength of these studies, it is important to keep in mind that not all the changes in signal at these loci are solely due to consumption of cigarettes or alcohol, and it is possible that in infrequent cases, rare genetic variation or exposure to as of yet uncharacterized xenobiotics/pharmaceuticals may also alter methylation at these loci. Nevertheless, by themselves, these two markers are strong predictors of mortality. Their more modest reduced hazard ratio, in particular for cg05575921, in the full model is indicative of the large extent to which the biology captured by these metrics also map smoking associated diseases such as COPD [36]. 

In contrast, establishing interpretability for the LEA is difficult because of the complexity of the marker set. In this study, we show that even after the effects of age, COPD, CHD, dementia, stroke, smoking, and drinking are addressed, a small portion of the variance for predicting survival still remains. What else is driving those still significant changes in methylation is not clear and the pathway to understanding those changes is uncertain. This is not the case for the epigenetic markers of substance use where our understanding of the relationship of these variables to key clinical outcomes in this communication is guided by decades of studies that used older types of biomarkers for smoking and drinking, such as cotinine and carbohydrate deficient transferase [37,38]. Still, two approaches to understand the independent effects of LEA may be useful. The first is a focus on changes at markers that are not known to be affected by smoking or drinking while the second is to control all analyses for methylation at the two substance biomarker loci. Still, no matter what approach is used, establishing what additional risk information that LEA may be capturing and to which diseases that signal may map will be challenging because of the scarcity of high quality epigenetic data sets, the sensitive nature of many of these key outcome variables and the use of self-report to assess affected status for key health outcomes (e.g., in this Wave 8 data set, self-report of COPD was used).

For those investigators who have DNA and wish to know which of these tools to use, to a certain extent the measure one uses depends on the question one wishes to answer and the material with which one has to work. If one only has DNA and no other knowledge, the LEA may be the most useful tool to use because of the ability of this metric to calculate age, the major predictor of mortality. This ability is of particular interest to those in forensics [39]. However, if one has age and gender information already, which will be the case in the vast majority of scientific studies, the substance use indices by themselves appear to be better able to predict mortality. What is more, they also convey an understanding of preventable mechanisms (i.e., smoking and drinking) of disease. This information is of critical importance to clinicians striving to target discrete medical or psychosocial variables in either treatment or prevention programs. The addition of LEA adds additional information with respect to mortality to these two metrics. But we believe that generating actionable mechanistic interpretations of the results will be challenging and stress the need for additional, publicly available data sets for individuals of all ancestries to help identify other potentially preventable causes of morbidity and mortality.

## 5. Conclusions

The take home message from this study for those interested in public health is clear. In contrast by what may be indicated by some and consistent with the work of Brenner and others, smoking and drinking are significant causes of mortality in the elderly. Because these substance use syndromes are inherently treatable, efforts to combat these two substance use syndromes may lead to considerable improvements in the well-being of our elderly. Interpretations that suggest otherwise detract from the important work of public health officials in improving healthy living in our elderly.

## Figures and Tables

**Figure 1 genes-10-00051-f001:**
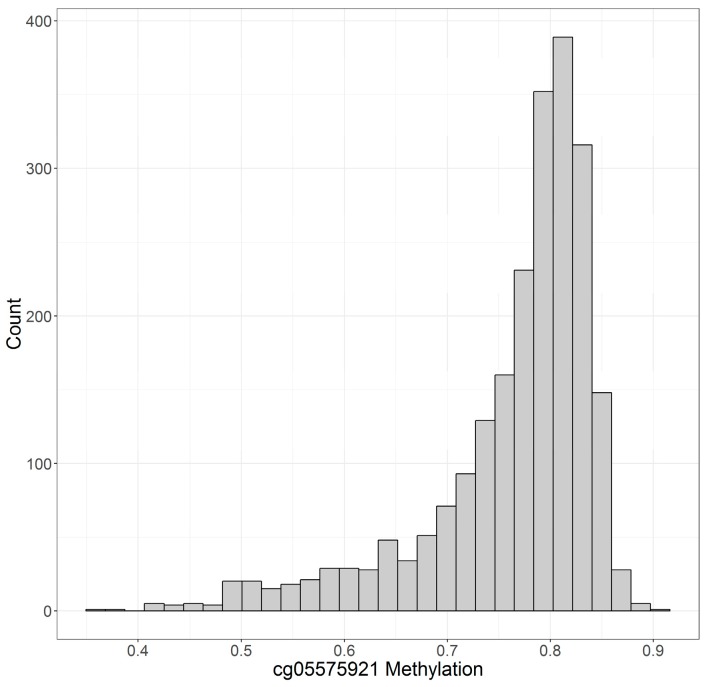
The distribution of cg05575921 methylation in the Framingham Heart Study (FHS) Offspring Cohort. Methylation at this locus is expressed as fractional methylation. *N* = 2256.

**Figure 2 genes-10-00051-f002:**
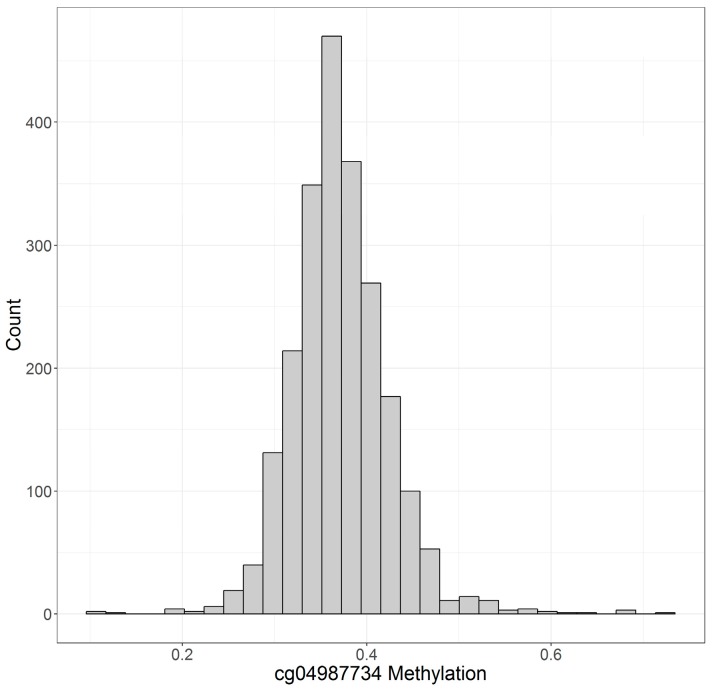
The distribution of cg04987734 methylation. Methylation at this locus is expressed as fractional methylation. *N* = 2256.

**Figure 3 genes-10-00051-f003:**
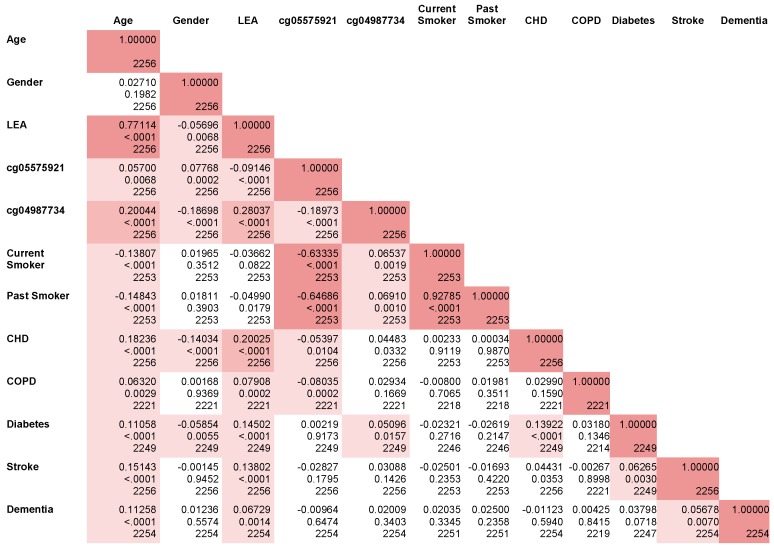
Pearson correlation coefficient of key variables.

**Table 1 genes-10-00051-t001:** Key demographic and clinical characteristics of participants.

	All	Male	Female
**Number of Participants**	2256	1022	1234
**Age at Intake ^†^**	66.3 ± 8.9 years	66.1 ± 8.8 years	66.5 ± 9.0 years
**Current Smoking Status ^‡^**			
Yes	179 (8.0)	75 (7.3)	104 (8.4)
No	2074 (91.9)	944 (92.4)	1130 (91.6)
Missing	3 (0.1)	3 (0.3)	0 (0.0)
**Past Smoking Status**			
Yes	203 (9.0)	86 (8.4)	117 (9.5)
No	2050 (90.9)	933 (91.3)	1117 (90.5)
Missing	3 (0.1)	3 (0.3)	0 (0.0)
**CHD**			
Yes	322 (14.3)	201 (19.7)	121 (9.8)
No	1934 (85.7)	821 (80.3)	1113 (90.2)
**COPD**			
Yes	47 (2.1)	21 (2.1)	26 (2.1)
No	2174 (96.4)	984 (96.3)	1190 (96.4)
Missing	35 (1.5)	17 (1.6)	18 (1.5)
**Diabetes**			
Yes	271 (12.0)	144 (14.1)	127 (10.3)
No	1978 (87.7)	874 (85.5)	1104 (89.5)
Missing	7 (0.3)	4 (0.4)	3 (0.2)
**Stroke**			
Yes	103 (4.6)	47 (4.6)	56 (4.5)
No	2153 (95.4)	975 (95.4)	1178 (95.5)
**Dementia**			
Present	10 (0.4)	3 (0.3)	7 (0.6)
Maybe	18 (0.8)	9 (0.9)	9 (0.7)
None	2226 (98.7)	1009 (98.7)	1217 (98.6)
Missing	2 (0.1)	1 (0.1)	1 (0.1)
**LEA**	58.8 ± 9.4 years	59.4 ± 9.5 years	58.3 ± 9.4 years
**Average Methylation**			
cg05575921	76.4 ± 8.4%	75.7 ± 9.0%	77.0 ± 7.8%
cg04987734	37.1 ± 5.2%	38.2 ± 5.0%	36.2 ± 5.2%

† Mean ± Standard Deviation for Continuous Measures; ‡ *N* (%) for Categorical Measures. CHD, coronary heart disease; COPD, chronic obstructive pulmonary disease status; LEA, Levine EA index.

**Table 2 genes-10-00051-t002:** Univariate associations of clinical and epigenetic characteristics with mortality.

Predictor	HR (95% CI)	*N* (Events)
**Age at Intake ^†^**	2.59 (2.28, 2.93) ***	2256 (288)
**Sex**		
Male vs. Female	1.50 (1.19, 1.89) **	2256 (288)
**CHD**		
Yes vs. No	3.10 (2.42, 3.98) ***	2256 (288)
**COPD**		
Yes vs. No	4.74 (3.07, 7.33)***	2221 (284)
**Diabetes**		
Yes vs. No	2.22 (1.67, 2.93) ***	2249 (286)
**Stroke**		
Yes vs. No	4.77 (3.42, 6.65) ***	2256 (288)
**Dementia**		
Present vs. None	5.83 (2.59, 13.11) ***	2254 (286)
**LEA ^†^**	2.34 (2.12, 2.59) ***	2256 (288)
**Average Methylation ^†^**		
cg05575921	0.70 (0.64, 0.76) ***	2256 (288)
cg04987734	1.53 (1.40, 1.66) ***	2256 (288)

† Continuous measures are standardized; * *p* < 0.05, ** *p* < 0.01, *** *p* < 0.0001. HR; Hazard Ratio; CI, confidence interval.

**Table 3 genes-10-00051-t003:** Prediction accuracy (Harrell’s C), strength of association (Pseudo R^2^), integrated discrimination index (IDI) and net reclassification improvement (NRI) for multivariate Cox proportional hazards models.

Model ^†^	Predictors	Harrell’s C	Pseudo R^2^	IDI (95% CI)	NRI (95% CI)
**1**	**Age, Sex**	**0.742**	**0.105**	-	-
2	Model 1 + CHD	0.755	0.113	0.0052 (−0.0011, 0.0201)	0.157 (−0.0283, 0.261)
3	Model 1 + COPD	0.752	0.116	0.0164 (0.0041, 0.0424)	0.0784 (−0.168, 0.136)
4	Model 1 + Diabetes	0.748	0.109	0.0002 (−0.0033, 0.0062)	0.0820 (−0.0140, 0.150)
5	Model 1 + Stroke	0.753	0.116	0.0157 (0.0021, 0.0361)	0.0313 (−0.163, 0.145)
6	Model 1 + Dementia	0.744	0.108	0.0018 (−0.0028, 0.0156)	−0.0590 (−0.249, 0.0404)
7	Model 1 + LEA	0.760	0.119	0.0149 (0.0055, 0.0312)	0.272 (0.160, 0.343)
8	Model 1 + cg04987734	0.754	0.114	0.0129 (0.0007, 0.0304)	0.141 (−0.0409, 0.226)
9	Model 1 + cg05575921	0.774	0.130	0.0290 (0.0117, 0.0527)	0.237 (0.147, 0.320)
10	Model 1 + cg04987734, cg05575921	0.779	0.135	0.0391 (0.0168, 0.0788)	0.264 (0.152, 0.349)
11	Model 1 + LEA, cg04987734, cg05575921	0.787	0.142	0.0473 (0.0249, 0.0771)	0.313 (0.204, 0.396)
12	Model 1 + CHD, COPD, Diabetes, Stroke, Dementia	0.779	0.139	0.0352 (0.0151, 0.0711)	0.277 (0.152, 0.374)
13	Model 1 + CHD, COPD, Diabetes, Stroke, Dementia, LEA	0.788	0.151	0.0578 (0.0331, 0.0980)	0.286 (0.195, 0.390)
14	Model 1 + CHD, COPD, Diabetes, Stroke, Dementia, cg05575921	0.801	0.161	0.0681 (0.0394, 0.117)	0.326 (0.262, 0.417)
15	Model 1 + CHD, COPD, Diabetes, Stroke, Dementia, cg04987734, cg05575921	0.806	0.167	0.0836 (0.0528, 0.138)	0.329 (0.257, 0.435)
16	Model 1 + CHD, COPD, Diabetes, Stroke, Dementia, LEA, cg04987734, cg05575921	0.810	0.173	0.0946 (0.0603, 0.145)	0.358 (0.271, 0.451)

† Missing values for predictors result in analysis sample size of 2212 participants (280 events) for all models.

**Table 4 genes-10-00051-t004:** Multivariate associations of clinical and epigenetic characteristics with mortality.

Predictors	z-Value	HR (95% CI)
**Age at Intake ^†^**	5.77	1.72 (1.43, 2.07) ***
**Sex**		
Male vs. Female	1.43	1.20 (0.94, 1.53)
**CHD**		
Yes vs. No	4.47	1.85 (1.41, 2.42) ***
**COPD**		
Yes vs. No	5.20	3.25 (2.09, 5.07) ***
**Diabetes**		
Yes vs. No	1.60	1.27 (0.95, 1.71)
**Stroke**		
Yes vs. No	6.38	3.10 (2.19, 4.39) ***
**Dementia**		
Present vs. None	2.90	3.37 (1.48, 7.69) **
**LEA ^†^**	4.01	1.44 (1.20, 1.72) ***
**Average Methylation ^†^**		
cg05575921	−6.45	0.71 (0.64, 0.79) ***
cg04987734	3.83	1.21 (1.10, 1.34) **

^†^ Continuous measures are standardized; * *p* < 0.05, ** *p* < 0.01, *** *p* < 0.0001.

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
