# Peer review of "A Direct Comparison of the Relationship of Epigenetic Aging and Epigenetic Substance Consumption Markers to Mortality in the Framingham Heart Study"

_genes, 2019, doi:10.3390/genes10010051_

Round 1
Reviewer 1 Report
Mills and colleagues did a statistical post analysis on the Framingham Heart Study collective. They confirm several indices for Epigenetic Aging and describe the potency of two discrete loci for mortality influenced by either smoking(cg05575921) and drinking alcohol(cg04987734). Especially the discussion is very detailed and discusses the relevant points in question while highlighting the applicability and limitations of the results.
The manuscript is of very high quality, both from language and statistics standpoint.
Minor comments:
1. Dementia has a very low percentage in the collective investigated. This could result in an unnecessarily high bias towards the results. It would be helpful to see the Model results (Models 12-16) without this influence in the supplementary material and commenting on the differences in the results section.
2. Since both smoking and alcohol abuse have clear correlations to cancer development, it would seem helpful to perhaps do an isolated analysis on cancer risk assessment and reconfirmation of the predictive capacity of the Biomarker positions in question. If no correlation is observed, this should also clearly be stated in the results.
Author Response
Comment: Dementia has a very….. Helpful to see the Model Results without dementia…….
The Reviewer is quite correct about the scarcity of dementia in the sample. As such, one would expect little impact of including or excluding the data. To test that hypothesis, we re-ran models 12 through 16 while excluding the dementia data. Not surprisingly, this exclusion had a negligible effect on the outcomes. To better inform the reader, we put a statement to this effect at the end (line 246) of this page and have pasted the results into the bottom of this response (see the table pasted down below). I think the reviewer will agree that the results are relatively unchanged and that including the Table may not be all that useful to the reader. Still, we would also be happy to put this in as a supplementary table but do not believe that anyone would click through to see it. However, if the Reviewer feels otherwise, we are perfectly happy to do so. It would take only a minute or two of our time.
Comment: Smoking and alcohol abuse have clear correlations to cancer……it would seem helpful to to an isolated analysis on cancer risk.
The reviewer is absolutely correct. However, three years ago when we asked for this data, we did not get the clinical information with respect to cancer diagnosis because at the time, it was not relevant to our studies. To get that information, we would have to resubmit our IRB application (Framingham requires IRB approval to get access for the data). Then, we would have to petition the study to get the information- this step alone would take up to six months. Finally, then we would have to figure out how to classify the various types of cancers before we can analyze the data. It is an important question but it may be beyond the scope of the current study and our capabilities as investigators. Indeed, in a recent NIH grant review, our colleagues pointed out to us that I nor anyone was not a Cancer expert and we needed help. Needless to say, we are adding a lung cancer expert to the study. At the same time, others may be better suited to parsing this out...if they can get the data from Framingham!
Model† | Predictors | Harrell’s C | Pseudo R2 | IDI | NRI |
1 | Age, Sex | 0.742 | 0.105 | -- | -- |
2 | Model 1 + CHD | 0.755 | 0.113 | 0.0052 | 0.157 |
3 | Model 1 + COPD | 0.752 | 0.116 | 0.0164 | 0.0784 |
4 | Model 1 + Diabetes | 0.748 | 0.109 | 0.0002 | 0.0820 |
5 | Model 1 + Stroke | 0.753 | 0.116 | 0.0157 | 0.0313 |
6 | Model 1 + Dementia | 0.744 | 0.108 | 0.0018 | -0.0590 |
7 | Model 1 + LEA | 0.760 | 0.119 | 0.0149 | 0.272 |
8 | Model 1 + cg04987734 | 0.754 | 0.114 | 0.0129 | 0.141 |
9 | Model 1 + cg05575921 | 0.774 | 0.130 | 0.0290 | 0.237 |
10 | Model 1 + cg04987734, cg05575921 | 0.779 | 0.135 | 0.0391 | 0.264 |
11 | Model 1 + LEA, cg04987734, cg05575921 | 0.787 | 0.142 | 0.0473 | 0.313 |
12 | Model 1 + CHD, COPD, Diabetes, Stroke | 0.777 | 0.136 | 0.0367 | 0.274 |
13 | Model 1 + CHD, COPD, Diabetes, Stroke, LEA | 0.787 | 0.148 | 0.0585 | 0.264 |
14 | Model 1 + CHD, COPD, Diabetes, Stroke, cg05575921 | 0.800 | 0.159 | 0.0703 | 0.341 |
15 | Model 1 + CHD, COPD, Diabetes, Stroke, cg04987734, cg05575921 | 0.805 | 0.164 | 0.0851 | 0.326 |
16 | Model 1 + CHD, COPD, Diabetes, Stroke, LEA, cg04987734, cg05575921 | 0.809 | 0.170 | 0.0952 | 0.361 |
†Missing values for predictors result in analysis sample size of 2212 participants (280 events) for all models
Reviewer 2 Report
The study presented is very interesting especially for considering the epigenetic index of alcohol consumption cg04987734, other than the epigenetic marker of cigarette consumption and the levine EA clock, in relation with survival in the FHS Offspring Cohort. The methods employed and results are adequately described.
The statistical models applied, especially model 16 (who takes into account all the variables), give good results, but, have the authors considered eventual professional exposure to xenobiotics in the analyzed cohort? Exposure to certain chemicals could be also responsible for alterated methylation of the considered markers?
Table 5 is not present in the manuscript.
The caption of Figure 3 is not present.
In the abstract it is reported "cg04987732" at line 29, would the authors mean cg04987734?
The same is at page 10, line 281.
Author Response
Comment: Have the authors considered …exposure to xenobiotics…could be responsible for altered methylation?
Response: No biomarker is perfect. And indeed, we are aware of individuals who seem to have smoked but do not seem to have de-methylated at cg05575921 in response to smoking. We have their DNA and sera collected and are planning to explore this using our other tools to better understand this observation. At the same time, it is certainly probable that exposure to certain pesticides or pharmaceuticals could result in demethylation of cg05575921. We just haven't observed that as of yet. But as Reviewer knows absence of evidence is not evidence of absence. We have put a comment on to page 10, line 331 to add caution to interpretations with respect to the specificity of the findings. Here is the comment: “Still, despite the strength of these studies, it is important to keep in mind that not all the changes in signal at these loci are solely due to consumption of cigarettes or alcohol, and it is possible that in infrequent cases, rare genetic variation or exposure to as of yet uncharacterized xenobiotics/pharmaceuticals may also alter methylation at these loci.”
Comment: Table 5 is not present in the manuscript.
Response: Table 5 is actually Figure 3. To make a long story short, because of the size restrictions for Tables that prevented me from presenting the correlations in Figure 3 as a Table (was just too wide), I elected to present the correlations as a Figure. Figure 3 depicts those correlations. There is no Table 5.
Comment: The caption of Figure 3 is not present.
This is related to the above comment. The caption now is on Page 9, line 275 and was erroneously labeled by me as Table 5. There is no Table 5.
Comment: In the abstract it is reported "cg04987732" at line 29, would the authors mean cg04987734? The same is at page 10, line 281.
That is due to sloppiness on my part (I was rather fatigued when I re-wrote the abstract late at night). I have corrected the typos.